# The Role of Body Image Perception on a Continuum from Dysfunctional to Healthy Eating Attitudes and Behaviors Among People Seeking Treatment

**DOI:** 10.3390/jcm14020597

**Published:** 2025-01-17

**Authors:** Francisca Bourbeau, Stéphane Bouchard, Giulia Corno, Johana Monthuy-Blanc

**Affiliations:** 1Loricorps Research Unity, Research Center of Mental Health University Institute of Montreal (CR-IUSMM), 7331, Rue Hochelaga, Montreal, QC H1N 3V2, Canada; francisca.bourbeau@uqtr.ca (F.B.); stephane.bouchard@uqo.ca (S.B.); giulia.corno@ssss.gouv.qc.ca (G.C.); 2Département d’Anatomie, Université du Québec à Trois-Rivières, 3351 Boulevard des Forges, Trois-Rivières, QC G8Z 4M3, Canada; 3Cyberpsychology Laboratory, Université du Québec en Outaouais, 283, Boulevard Alexandre-Taché, Gatineau, QC J8X 3X7, Canada; 4Département de Psychoéducation et de Psychologie, Université du Québec en Outaouais, 283, Boulevard Alexandre-Taché, Gatineau, QC J8X 3X7, Canada; 5Centre Intégré de Santé et des Services Sociaux de l’Outaouais, Gatineau, QC J8P 7H2, Canada; 6Département de Sciences de l’Éducation, Université du Québec à Trois-Rivières, 3351 Boulevard des Forges, Trois-Rivières, QC G8Z 4M3, Canada

**Keywords:** body image perception, eating and attitudes and behaviors, virtual reality

## Abstract

**Background:** Body image disturbance has been associated with various health conditions and has the potential to trigger the development of unhealthy behaviors, including dysfunctional attitudes and eating behaviors, which may evolve into eating disorders. This study explores the relationship between body image variables—such as global self-worth, physical self-worth, and physical attractiveness—and the continuum of eating attitudes and behaviors. **Methods:** A canonical correlation analysis was performed to assess the multivariate relationship between attitudinal and perceptual variables and the continuum of eating behaviors, with a total of 113 cases analyzed. **Results:** The findings indicate that intuitive eating and a positive body image (including global self-worth and physical attractiveness) are most strongly associated with the functional end of the continuum, while disordered eating behaviors and body dissatisfaction are linked to the dysfunctional end. **Conclusions:** These results suggest that interventions targeting the emotional and attitudinal dimensions of body dissatisfaction, whether delivered in vivo or via virtual reality (e.g., weight exposure), may facilitate a shift toward healthier, more functional eating behaviors along the continuum.

## 1. Introduction

Post-pandemic figures indicate that over 1,800,000 Canadians are affected by obesity, with 15% of them experiencing an eating disorder that does not meet the strict criteria for a mental health diagnosis [1].

Both an increase in and a broader range of disordered eating and eating disorders have been observed [2,3]. For example, over 53% of participants reported engaging in unhealthy weight control behaviors, and over 14% reported binge eating episodes in the past month [4]. Disordered eating refers to a wide range of eating behaviors that do not meet the strict clinical criteria for an eating disorder (formally defined by the Diagnostic and Statistical Manual of Mental Disorders). Disordered eating behaviors are often understudied and may go unnoticed [5]. Research continues to show that a much larger proportion of the population than previously thought is vulnerable to eating health problems, and these findings currently reaffirm the importance of considering eating attitudes and behaviors (EABs) as a model that includes both ED and disordered eating. Up to 48% of individuals exhibit disordered eating and eating disorders when conceptualized along a continuum of eating attitudes and behaviors (EABs) [6]. EABs are conceptualized along a continuum, with one pole corresponding to functional EABs and the other to dysfunctional EABs. Dysfunctional EABs are characterized by behaviors such as food restriction, binge eating and emotional eating, and include disordered eating in community populations and eating disorders in clinical populations [7,8,9]. The opposite end reflects more functional behaviors associated with intuitive eating. Intuitive eating is a dynamic process that integrates mind–body–food harmony by attuning and responding to the body’s signals to meet both physical and psychological needs [10]. These two poles are characterized by distinct behaviors, and previous studies have typically measured one or the other, but not both at the same time. Between the two poles lies a continuum in which domains are dominated by the role of different variables. A key point on this continuum is represented by the contribution of body image perceptions. Recognition of this continuum implies more than simply focusing on the range between low and high clinical severity. The absence of dysfunctional beliefs, attitudes and behaviors does not provide sufficient information about the presence of healthy and resilient beliefs, attitudes, and behaviors. This is similar to the difference between the absence of illness and health, where being healthy means more than just not being sick [11]. Negative body image perceptions tend to trigger and induce dysfunctional EABs, whereas positive body image perceptions promote functional EABs. Classically, body image perceptions refer to the cognitive-behavioral model proposed by Cash (2012) [12]. In this model, body image perceptions include an emotional dimension (e.g., body satisfaction or dissatisfaction), a perceptual dimension (e.g., body distortions), a behavioral dimension (e.g., actions taken to modify body shape), and an attitudinal dimension (e.g., itself composed of cognitive and affective dimensions). Recent findings have also concluded that the attitudinal dimension refers to physical attractiveness as the perceived ability to maintain an attractive body over time [13]. According to Fox and Corbin’s (1989) model, physical attractiveness is a component of physical self-worth, which refers to the overall sense of pride in one’s physical abilities and appearance [14]. This physical self-worth is influenced by several factors and is ultimately included in global self-esteem [15,16].

Negative body image perceptions are a key factor that precedes and causes dysfunctional EABs. Specifically, some studies show that individuals with dysfunctional EABs perceive body attractiveness differently than the general population. For example, they tend to rate the bodies of others with a healthy weight as more overweight, view underweight women as normal weight (and more attractive), and spend significantly more time focusing on unattractive body parts after a stimulus presentation [17,18]. This heightened sensitivity to psychological and perceptual factors may contribute to increased social comparisons, especially with idealized body standards, that promote feelings of unattractiveness [17,18]. These comparisons, in turn, often drive dysfunctional eating behaviors aimed at weight control [17].

In addition, individuals with dysfunctional EABs often experience low global self-esteem, characterized by persistent negative thoughts and feelings about themselves [19]. Meta-analyses and longitudinal studies highlight a vicious cycle in which low self-esteem increases vulnerability to developing severe dysfunctional EABs, while the presence of dysfunctional EABs further erodes self-esteem [19]. In addition, studies have shown that individuals with dysfunctional EABs often have distorted perceptions of their body size and shape and body dissatisfaction [20]. Furthermore, the severity of this distortion serves as a negative prognostic factor for long-term outcomes and reinforces dysfunctional eating attitudes and behaviors [21]. Body dissatisfaction, in turn, promotes the adoption of maladaptive eating behaviors aimed at controlling weight and altering physical appearance [22]. Significant correlations have been found between body dissatisfaction and several dysfunctional EAB manifestations, including binge eating and fasting. Studies also identify body dissatisfaction as the strongest predictor of the onset of dysfunctional EABs, particularly among adolescents at risk for more severe dysfunctional EABs [23,24].

The assessment of body image disturbances (e.g., body dissatisfaction) can be evaluated from two visual perspective frames of one’s own body: the allocentric and the egocentric perspective [25,26,27,28]. In the egocentric perspective, also referred to as the first-person perspective, the virtual body is seen through one’s own eyes and is perceived as a feeling of experiencing the body based on sensory input [26,28]. In the allocentric perspective, also referred to as the third-person perspective, the virtual body is perceived as a representation of our body observed in the environment [26,28]. Results from several studies conducted with community and clinical samples highlight a clear distinction between the two perspectives. In addition, these studies show that they complement each other by addressing either the attitudinal or the perceptual aspects of body image perception. The attitudinal, cognitive, and affective dimensions are most often associated with the allocentric perspective [25]. In contrast, the perceptual and affective dimensions of body image are more related to the egocentric perspective [25,27,29]. In summary, individuals tend to perceive their bodies as objects (non-embodied and through interpersonal comparison) from an allocentric perspective, whereas they experience their bodies as subjects (embodied and through intrapersonal comparison) from an egocentric perspective [27]. A recent study also suggests that body dissatisfaction, both allocentric and egocentric, cannot be explained solely by attitudinal dimensions of body image perceptions, but also by broader elements such as global and physical self-worth [30]. Riva and colleagues investigated how virtual reality can modify allocentric body memory in the severe form of dysfunctional EAB populations. The allocentric lock theory suggests that individuals with anorexia rely on an observer-based self-image and past memories to evaluate their appearance. Disruptions in bodily perception integration contribute to dysfunctional eating behaviors [31].

In summary, the literature is dominated by studies that focus on one of the two poles (i.e., predicting scores on dysfunctional or functional measures). Therefore, further research is imperative to gain a comprehensive understanding of the dynamics between attitudinal and perceptual body image variables and eating behaviors. Research should move beyond the traditional narrow focus on eating attitudes and behaviors and measure their full range, from illness to health.

The main objective of this article is to investigate the relationships between attitudinal and perceptual body image variables, global self-worth, physical self-worth, and physical attractiveness in relation to the continuum of EABs, as illustrated in Figure 1. This research aims to identify specific attitudinal body image facets that often co-occur with different eating attitudes and behaviors, with the potential to inform more effective interventions for eating behaviors. Given the established relationship between low self-esteem, body dissatisfaction, DEAB, and body dissatisfaction with low intuitive eating, we hypothesized that body dissatisfaction and global self-worth would significantly correlate with the continuum of eating attitudes and behaviors [19,32].

## 2. Materials and Methods

### 2.1. Data Sources

This retrospective study was conducted by analyzing the anonymized secondary data collected in the Loricorps database. The Loricorps database (LDB), a secure data platform used in this study, comprises data collected during enrollment in the *e*LoriCorps program 1.0, an e-Health transdisciplinary program, offered at a university-based clinic, between September 2017 and December 2022 [33]. The *e*LoriCorps program 1.0 addresses dysfunctional eating attitudes and behaviors resulting from perceptual disorders of all ages and forms, sub-clinical or clinical (eating disorders, sports eating disorders, etc.), of mild to moderate severity. Exclusion criteria were the presence of a severe self-reported comorbid psychiatric condition (e.g., personality disorders, psychosis or severe anxiety or depression). The eating disorder-related data were collected by healthcare providers during sessions conducted in accordance with the intervention protocol of the *e*LoriCorps program 1.0. Comorbidities were assessed by the *e*LoriCorps Program 1.0 transdisciplinary clinical team and confirmed by a diagnostic specialist (primarily physician or psychologist) using DSM-5 criteria.

### 2.2. Procedures

Participants provided written informed consent and allowed the use of their anonymous data in accordance with current legislation regarding the protection of personal data (Declaration of Helsinki of 1975, as revised in 2018, and the Tri-Council Policy Statement: Ethical Conduct for Research Involving Human Subjects—TCPS 2 of 2018). This study was approved by the Ethics Committee of the Université du Québec à Trois-Rivières (Quebec, Canada; reference number: CER-22-293-10.02; 2 November 2023).

### 2.3. Sample

The sample consisted of 103 women and 10 males. Ages ranged from 13 to 80 years (M = 37.11) and Body Mass Index (BMI, kg/m^2^) from 11 kg/m^2^ to 63 kg/m^2^ (M = 31.91, SD = 10.61).

### 2.4. Assessment Measures

*Eating Disorder Examination Questionnaire* (EDE-Q; [34]). The global score is obtained by calculating the average of the scores from the 4 subscales (shape concerns, eating concerns, weight concerns and restraint). Participants rated the items on a rating scale ranging from 0 (“no days”) to 6 (“every day”). The total score for each subscale was obtained by calculating the average of all responses. In this study, the EDE-Q showed a good internal consistency for the total (Cronbach’s α = 0.885).

*Intuitive Eating Scale—2* (IES-2; [35]). The total score of the IES-2 is calculated by summing the scores for each subscale. For each subscale, the sum is then divided by the number of items in that subscale. The participant responds to the items using a Likert-type response scale ranging from 1 “strongly disagree” to 5 “strongly agree”. In this study, the EDE-Q showed a good internal consistency for the total scale (Cronbach’s α = 0.723).

*eLoriCorps-Immersive Body Rating Scale version 1.1* (eLoriCorps-IBRS 1.1; [27]).

Participants were asked to self-report their age, while their current height and weight were reported by caregivers using the *e*LoriCorps 1.0 program to calculate BMI.

This virtual reality-based adaptation of paper-and-pencil figure rating scales was used to visually estimate body size from both allocentric and egocentric perspectives. It includes three virtual environments: a neutral environment, and two spatial frame perspectives (allocentric and egocentric), each with a male or female body continuum. First, the neutral environment allowed participants to develop the navigation skills necessary to move within the virtual environment and around identical tubular structures, without the presence of virtual bodies. Participants were then asked, from both the allocentric and egocentric perspectives, to select the virtual body that most closely represented their ideal and perceived body size. In the allocentric perspective, users viewed a lineup of seven increasing BMIs, ranging from 15 to 33 kg/m^2^, or nine virtual bodies (with increasing BMIs ranging from 15 to 40 kg/m^2^). All virtual bodies were visible in the user’s field of vision. To select their perceived and ideal body size, participants walked toward the virtual body that best reflected their perceived and ideal body size. In the egocentric perspective, users were told that they would experience the virtual body as if it were their own, looking down from the chest. The user could move their head along three degrees of freedom (yaw, pitch, and roll). The experimenter then explained that users would experience each virtual body, from the thinnest to the largest. To select their perceived and ideal body size, participants guided the experimenter to the virtual body that most closely matched their ideal and perceived size. Participants were asked to select the virtual body that closely represented their ideal and perceived body size. Z-scores were calculated for each variable measured due to the use of different scales (i.e., a 7-point and a 9-point scale), including Z ideal body size—Allocentric, Z Ideal body size Egocentric, Z Perceived body size Egocentric, Z Perceived body size Allocentric (For a detailed description of the procedure please refer to Monthuy-Blanc et al., 2022 [27]). Visual-perceptual body dissatisfaction refers to the difference between perceived and ideal body size. A score different than 0 indicates body dissatisfaction with one’s body. A positive score indicates that a participant’s ideal body size was thinner that their perceived body size, while a negative score suggests that a participant’s ideal body size was bigger than their perceived body size. Z-scores were computed for egocentric (i.e., Z VPBD-Egocentric) and allocentric (i.e., Z VPBD-Allocentric) visual-perceptual body dissatisfaction.

*Physical Self-Inventory* (PSI; [36]). The global self-worth, physical self-worth and physical attractiveness subscales of the PSI were used to evaluate participants’ global self-worth, physical self-worth and physical attractiveness according to the model of Fox and Corbin (1989) [14]. Answers are rated on a scale ranging from 1 (“not at all”) to 6 (“Absolutely”). In the present study, internal consistency varied between subscales (global self-worth: Cronbach’s α = 0.566; physical self-worth: Cronbach’s α = 0.258; physical attractiveness: Cronbach’s α = 0.469).

*Eating Disorder Inventory—very short version* (EDI-VS; [37]). The symptoms index score of the EDI was calculated by summing the scores of the items associated with three subscales (bulimia, body dissatisfaction, and drive for thinness). Answers are rated on a scale ranging from 0 (“not at all”) to 5 (“extremely”). In the present study, internal consistency for the index symptoms is insufficient (Cronbach’s α = 0.556).

### 2.5. Descriptive Statistics

This study used a canonical correlation analysis [38], using the SPSS version 29 canonical correlation command and MANOVA command with the discriminate option. This analysis follows an exploratory approach designed to determine the magnitude of the relationship between two multivariate sets of variables referred to as canonical variables. It directly examines the relationship between two sets of variables, which is more straightforward for understanding the interdependencies and avoids the need for specifying of the complex and restrictive model structure required in SEM [39,40,41]. The analysis created a canonical variable by weighting the measures of eating disorders and intuitive eating, and tested the statistical significance of the canonical correlation with the second canonical variable created by weighting attitudinal and perceptual variables such as egocentric body distortion, allocentric body distortion, egocentric body dissatisfaction, allocentric body dissatisfaction, symptom index of the Eating Disorder Inventory, global self-worth, physical self-worth and physical attractiveness. Factor loadings of the different measures within their respective canonical variates are reported with canonical loadings. Descriptive statistics on the main measures are reported in Table 1. To document the potential impact of sex on the results, all statistical analyses were also performed separately for females and males. The results did not differ when analyzed separately for each sex (i.e., significant differences remained significant and non-significant differences remained non-significant). Including age in the analysis also did not change the results, and the contribution of age was non-significant. Therefore, to maximize statistical power, results for sex and age are not reported (analyses by sex and with age are available upon request).

Analyses of the participants’ clinical Pearson’s bivariate correlations are reported in Table 2.

Canonical correlation analysis was used to assess the multivariate common relationship between attitudinal and perceptual variables and the eating behavior continuum. The analysis used these variables as predictors to assess the relationship between the two sets of variables. Two successive canonical correlation functions were obtained, corresponding to the number of variables in the smallest set of variables [42]. The first canonical function was statistically significant, with r1 = 0.780 (eigenvalue = 1.549, F = 8.805, *p* < 0.001; Wilk’s lambda = 0.353), and thus was the only one to be interpreted. In contrast, the second canonical function was not statistically significant (r2 = 0.318, eigenvalue = 0.112, F = 1.669, *p* = 0.125; Wilk’s lambda = 0.899). The first model demonstrated a statistically significant explanatory power, with an explained variance of the correlation of 93.23%, and 61% of the shared variance between the two canonical variable sets being explained. As shown in Table 3, both the Intuitive Eating Scale (IES) and the Eating Disorder Examination Questionnaire (EDE-Q-Total) significantly contributed to the canonical variates in Set 1. Specifically, a strong negative correlation was observed between intuitive eating and the canonical dimension, while the EDE-Q total score was positively associated with the canonical dimension.

The correlations between the variables measuring attitudinal and perceptual dimensions of body image and the canonical scores of the continuum from dysfunctional to healthy eating attitudes and behaviors (i.e., cross-loadings) were −0.18 (*p* = 0.06, ns) for body distortion from the egocentric perspective, −0.21 (*p* = 0.023) for body distortion from the allocentric perspective, 0.43 (*p* < 0.001) for body dissatisfaction from the egocentric perspective, 0.48 (*p* < 0.001) for body dissatisfaction from the allocentric perspective, 0.65 (*p* < 001) for the Symptom index of the EDI, −0.59 (*p* < 0.001) for the global self-worth subscale of the PSI, −0.45 (*p* < 0.001) for the physical self-worth subscale of the PSI, and −0.54 (*p* < 0.001) for the physical attractiveness subscale of the PSI.

Figure 2 displays simultaneously the scatterplots of the two correlations between the canonical variate representing the set of attitudinal and perceptual variables (i.e., Set 2) and the weighted scores (i.e., after applying the derived canonical weights to the original variables) of the variables constituting the continuum from healthy (as measured by the IES) to dysfunctional (as measured by the EDEQ) eating. It shows the direction and strengths of the correlations, but most importantly that attitudinal and perceptual variables were contributing differently at both ends of the continuum. These results support the interest in conceptualizing EABs along a continuum from healthy to dysfunctional eating.

## 3. Discussion

The current article focuses on the relationships between attitudinal and perceptual body image variables, global self-worth, physical self-worth, and physical attractiveness and their contribution to the continuum from functional to dysfunctional EABs. Using a canonical regression analysis, we were able to create a latent construct representing the continuum from functional to dysfunctional eating attitudes and behaviors and found that all tested predictors were statistically significant, except for body distortion.

### 3.1. Integration of Key Concepts in Body Image Perception

Our findings highlight the importance of allocentric body dissatisfaction and perceived physical attractiveness in relation to dysfunctional EABs. Specifically, with regard to body dissatisfaction, these findings are consistent with current research that distinguishes between experiencing the body from an egocentric versus an allocentric perspective [27]. Overall, these findings lend support to both social comparison theory and objectification theory. Festinger’s (1954) social comparison theory posits that individuals tend to compare themselves to others in order to evaluate their own abilities and attributes [43]. The extant literature suggests that social comparison, whereby individuals evaluate their own appearance in comparison to those perceived as more attractive, is a significant driver of body dissatisfaction [44]. The tendency to view the body as an object to be improved and transformed is linked to self-objectification, a concept developed by Fredrickson and Roberts (1997) [45]. According to these authors, objectification theory explains how societal and media-driven beauty standards lead women to view themselves as objects to be evaluated based on appearance (allocentric perspective: “How do I look?”) rather than focusing on how they feel (egocentric perspective: “How do I feel?”). This objectified self-perception promotes dysfunctional eating behaviors as a way to cope with body dissatisfaction stemming from sociocultural pressures, such as emotional eating [46]. Ultimately, prolonged exposure to beauty ideals is likely to reinforce the notion of what constitutes an “attractive” body, increasing the risk of body dissatisfaction and dysfunctional EABs. In addition, research indicates that individuals with dysfunctional EABs tend to focus more intensely and frequently on body parts they deem unattractive on their own bodies [18].

Body dissatisfaction is a key factor in the development of eating disorders, while low self-esteem plays a central role in both the development and maintenance of body dissatisfaction [47,48]. Research suggests that global self-worth significantly influences EABs. According to Fairburn’s revised transdiagnostic cognitive-behavioral model (2003), low self-esteem is one of four mechanisms that sustain eating disorders [49]. Specifically, the overvaluation of weight and shape is perpetuated by low self-esteem, prompting individuals to regulate their weight and shape in order to enhance their sense of self-worth [49]. Distorted attitudes towards one’s actual and desired body can significantly influence various aspects of body image perception, including how individuals view their current body size, shape, and their idealized body [29]. Taken together, these findings underscore the importance of integrating key conceptual models of body image attitudes and perceptions.

### 3.2. To Unlock Body Dissatisfaction: Innovative Clinical Strategies

As previously mentioned, the population under study consists of individuals who wish to participate in a research intervention program. These participants exhibit negative affect and attitudes regarding body image perceptions, which have led to the crystallization of body dissatisfaction and the experience of disconnection from one’s body (disembodiment). In summary, the participants have already transitioned to the dysfunctional end of the spectrum, exhibiting a range of maladaptive eating attitudes and behaviors, including restraint, binge eating, and emotional eating. Several interventions have been identified as effective in reducing dysfunctional EABs and promoting functional EABs. These strategies are particularly effective when they target the affective and attitudinal dimensions of body dissatisfaction, thereby promoting overall eating wellness. In recent decades, advances in research and practice have led to the identification of promising approaches, not only in the understanding of body image issues but also in their evaluation and intervention. Cognitive-behavioral therapy (CBT) or culturally adapted cognitive-behavioral techniques that integrate culturally relevant values and norms, combined with innovative digital and technological interventions such as virtual reality, have shown great promise in understanding and addressing these challenges [50,51].

In virtual reality some interventions specifically target body dissatisfaction through virtual reality (e.g., the Perceptual Training Paradigm and the Allocated Intervention). Virtual reality offers a promising tool for gradual exposure to emotional discomforts, such as the fear of weight gain. Furthermore, it assists individuals in enhancing their awareness of body image and reframing maladaptive thoughts about weight. This process allows individuals to modify their perception of body size and re-evaluate the subjective distinctions between what is considered a “fat” or “thin” body [52,53,54,55].

In addition, various exercises can promote a more grounded, internal perspective of the body, emphasizing sensations and self-compassion rather than focusing on how one believes one is perceived externally (allocentric perspective). Cognitive behavioral therapy (CBT) has been demonstrated to be an efficacious approach for addressing body dissatisfaction. This is achieved by assisting individuals in reframing maladaptive thoughts about their bodies into more realistic and self-compassionate ones [55]. Self-compassion involves acknowledging one’s body-related stressors as an inherent aspect of the shared human experience, without an inclination to alter or evade these emotions [56]. This attitude of self-compassion can serve as a protective factor against dysfunctional coping strategies (e.g., restriction, binge eating) that are often used to escape unpleasant emotions [57]. In other words, individuals who activate a compassionate response and treat themselves with kindness when confronted with negative body image perceptions are less likely to resort to maladaptive coping mechanisms [58]. In conclusion, self-compassion interventions may enhance embodiment—fostering a positive connection with the body—while simultaneously reducing self-objectification [59].

Although self-compassion interventions have demonstrated efficacy in reducing shape concerns and self-objectification, further research is needed before clinical recommendations can be effectively formulated [58]. Furthermore, in stressful situations, combining strategies that address both food choices and stress management (such as cognitive interventions) may be promising tools for promoting healthier and more functional food choices [60]. Finally, it is important to gain a deeper understanding of the experience of disembodiment in individuals with dysfunctional EABs, as well as the strategies to promote embodiment in both virtual reality and in vivo settings.

## 4. Strengths and Limitations

To the best of our knowledge, no studies have focused on the role of body image variables (both allocentric and egocentric) in relation to eating behaviors, nor have they integrated these variables along the entire eating behaviors continuum from healthy to dysfunctional. This study could also provide insight into the specific attitudinal body image facets that most frequently occur alongside the eating attitudes and behaviors continuum, which may contribute to more targeted eating behaviors intervention. Nevertheless, given the sample, we find it valuable to report results from individuals with DEABs with mild to moderate severity. Our study provides information about a large portion of the population. As mentioned above, up to 48% of individuals may exhibit disordered eating and eating disorders when conceptualized along a continuum of eating attitudes and behaviors. In future research, it would be worthwhile to focus on more severe cases in order to better understand the relationship between eating behaviors and body image perceptions within this specific population. It should be noted, however, that the current study is not without limitations.

In terms of sample size, it would be valuable for future studies to examine the robustness of the associations and ensure better generalizability of the results by using a larger and more heterogeneous sample in terms of sex and gender. Given that the statistics only compared sexes and that the majority of the study sample consisted of women (with only 10 men), the findings may not be applicable to individuals of other genders, as body image perceptions can vary across genders, mainly, and sexes [61,62]. Future research should aim to include a more balanced representation of male and female samples to investigate potential gender-based differences. Indeed, it may be valuable to include certain sociodemographic variables that were not measured in the present study but could have an impact, such as sexual orientation, culture, or social support [63]. Furthermore, given that body dissatisfaction varies across different cultures, the inclusion of this variable in future research could provide additional insights and enhance the study’s relevance [51]. A culturomics approach could provide valuable insights into investigating EABs and the cultural representations of body image perceptions, spanning from micro to macro levels [64].

In addition, although the inclusion of a broad age range (13 to 80 years) can be informative, it may affect the generalizability of the results, particularly for less-represented age groups. Similarly, a wide range of BMIs, although valuable, may introduce bias in the interpretation of results related to eating behaviors and body image perceptions, as low BMIs may be more associated with restrictive eating behaviors, whereas higher BMIs may be more associated with binge eating [65,66]. Another limitation is the use of different rating scales in the *e*LoriCorps Immersive Body Rating Scale version 1.1, including a 7-point scale and a 9-point scale. Individuals with a high BMI may perceive themselves as larger than what the response scale accommodates, particularly in the 7-point scale, which may result in a significant response bias. Consequently, the restricted range of virtual bodies available on the *e*LoriCorps-IBRS (7-point scale) may limit the scope for participants to make a choice.

## 5. Conclusions

In conclusion, this study provides valuable insights into the complex relationships between body image perception, self-worth, and the continuum of eating attitudes and behaviors (EABs). The findings underscore the pivotal role of body image perceptions as a primary determinant of the EABs continuum, where alterations in these perceptions can significantly influence overall eating well-being. The empirical potential of CBT and virtual reality interventions offer promising avenues for positively reshaping body image perceptions. Further research is required to evaluate the efficacy of these novel approaches in guiding individuals toward the functional end of the EABs continuum, thereby providing new therapeutic opportunities for addressing dysfunctional EABs.

## Figures and Tables

**Figure 1 jcm-14-00597-f001:**
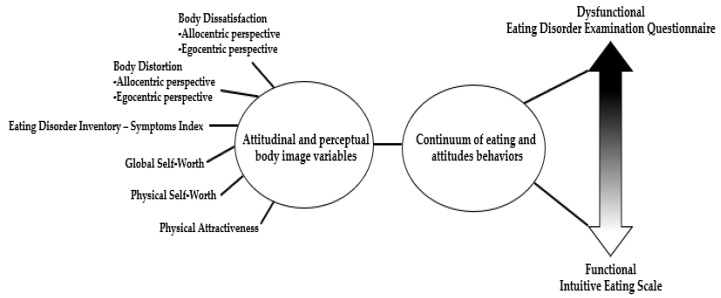
Relationship between body image variables, global self-worth, physical self-worth and physical attractiveness with the continuum of EABs.

**Figure 2 jcm-14-00597-f002:**
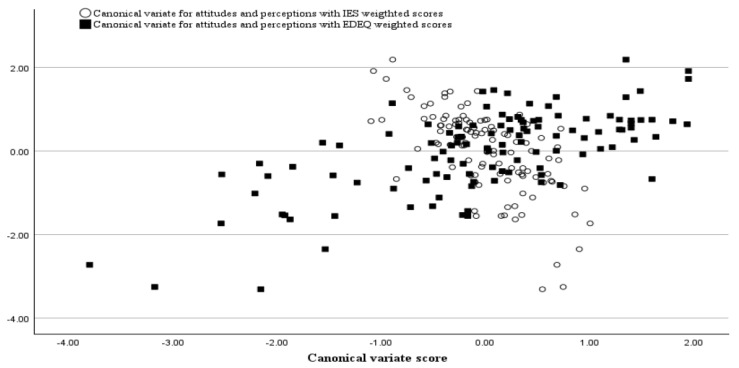
Scatterplot illustrating the correlations between the canonical variable representing the set of attitudinal and perceptual variables and weighted scores of the IES (empty circles) and of the EDEQ (black squares). Note: IES = Intuitive Eating Scale—total weighted score after applying the derived canonical function; EDEQ = Eating Disorder Examination Questionnaire—total weighted score after applying the derived canonical wight from the first canonical function.

**Table 1 jcm-14-00597-t001:** Descriptive statistics for body image and dysfunctional eating attitudes and behavior variables.

	M ± SD	Min–Max
IES-Total	2.51 ± 0.45	1.42–3.52
EDE-Q-Total	3.80 ± 1.15	0–5.75
*Body dissatisfaction*		
Allocentric perspective	−0.01 ± 0.86	−2.49–1.90
Egocentric perspective	−0.02 ± 0.89	−1.75–2.07
*Body distortion*		
Allocentric perspective	0.002 ± 0.97	−3.02–2.74
Egocentric perspective	0.02 ± 0.96	−3.37–2
EDI-SX	19.36 ± 5.42	2–30
PSI-GSW	2.16 ± 0.98	1–4.50
PSI-PSW	2.61 ± 1.20	1–6
PSI-PA	2.96 ± 1.05	1–5.50

Note. M: Mean; SD: Standard deviation; Min–Max: Minimum Maximum; IES-Total: Intuitive Eating Scale total score; EDE-Q-Total: Eating Disorder Examination Questionnaire Total score; EDI-SX: Eating Disorder Inventory—Symptom index; PSI-GSW: Physical Self Inventory—Global Self-Worth subscale; PSI-PSW: Physical Self Inventory—Physical Self-Worth; PSI-PA: Physical Self Inventory—Physical Attractiveness subscale.

**Table 2 jcm-14-00597-t002:** Participants’ clinical Pearson’s bivariate correlations among variables included in the canonical correlation analysis.

	1	2	3	4	5	6	7	8	9	10
1.IES-Total	-	−0.500	−0.178	0.218	0.410	−0.460	−0.514	0.503	0.479	0.414
2.EDE-Q-Total		-	−0.133	−0.156	0.341	0.375	0.607	−0.524	−0.314	−0.508
3. BDist_Ego			-	0.576	−0.424	−0.062	−0.053	0.228	0.311	0.177
4. BDist_Allo				-	−0.131	−0.239	−0.168	0.206	0.262	0.216
5. BDiss_Ego					-	0.638	0.270	−0.393	−0.451	−0.475
6. BDiss_Allo						-	0.346	−0.359	−0.402	−0.491
7. EDI-SX							-	−0.368	−0.241	−0.409
8. PSI-GSW								-	0.574	0.629
9. PSI-PSW									-	0.575
10. PSI-PA										-

Note. IES-Total: Intuitive Eating Scale Total score; EDE-Q-Total: Eating Disorder Examination Questionnaire Total score; BDist_Ego: Body Distortion in egocentric perspective from *e*LoriCorps-IBRS 1.1; BDist_Allo: Body Distortion in allocentric perspective from *e*LoriCorps-IBRS 1.1; BDiss_Ego: Body Dissatisfaction in egocentric perspective from *e*LoriCorps-IBRS 1.1; BDiss_Allo: Body Dissatisfaction in allocentric perspective from *e*LoriCorps-IBRS 1.1; EDI-SX: Eating Disorder Inventory—Symptom index; PSI-GSW: Physical Self Inventory—Global Self-Worth subscale; PSI-PSW: Physical Self Inventory—Physical Self-Worth; PSI-PA: Physical Self Inventory—Physical Attractiveness subscale.

**Table 3 jcm-14-00597-t003:** Canonical loadings.

Set 1 Canonical loadings		
Variable	1	2
IES-Total	−0.847 **	0.532
EDE-Q-Total	0.884 **	0.468
Set 2 Canonical loadings		
BDist-Ego	0.227	0.162
BDist-Allo	0.274	0.220
BDiss-Ego	−0.553 **	−0.266
BDiss-Allo	−0.614 **	−0.323
EDI-SX	−0.835 **	0.216
PSI-GSW	0.761 **	0.004
PSI-PSW	0.579 **	0.571
PSI-PA	0.687 **	−0.232

Note. ** *p* < 0.001, IES-Total: Intuitive Eating Scale Total score; EDE-Q-Total: Eating Disorder Examination Questionnaire Total score; BDist-Ego: Body Distortion in egocentric perspective from *e*LoriCorps-IBRS 1.1; BDist-Allo: Body Distortion in allocentric perspective from *e*LoriCorps-IBRS 1.1; BDiss-Ego: Body Dissatisfaction in egocentric perspective from *e*LoriCorps-IBRS 1.1; BDiss-Allo: Body Dissatisfaction in allocentric perspective from *e*LoriCorps-IBRS 1.1; EDI-SX: Eating Disorder Inventory—Symptom Index; PSI-GSW: Physical Self Inventory—Global Self-Worth Subscale; PSI-PSW: Physical Self Inventory—Physical Self-Worth; PSI-PA: Physical Self Inventory—Physical Attractiveness subscale.

## Data Availability

The data presented in this study are available on request from the corresponding author. The data are not publicly available due to ethical restrictions.

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
