# Peer review of "The Role of Body Image Perception on a Continuum from Dysfunctional to Healthy Eating Attitudes and Behaviors Among People Seeking Treatment"

_jcm, 2025, doi:10.3390/jcm14020597_

Round 1
Reviewer 1 Report
Comments and Suggestions for Authors
Overly Long and Complicated Title: A more concise title would facilitate a quicker understanding of the study focus.
The methodology explains that canonical correlation analysis (CCA) is chosen, but it does not explain why CCA was used rather than other methods, such as structural equation modelling (SEM). Explain your reason for this selection, and note any restrictions it may have.
Insufficient description of the VR intervention Please elaborate on the specific components of VR experiences (e.g., wearing extra weight, what are the targets of emotional and attitudinal aspects of body dissatisfaction)
The results section should contain specific statistical data (e.g., correlation coefficients, p-values, etc.) to adequately support the author's claims about the relationships between body image and eating behaviours.
Critically, the paper does not control for confounders such as age, gender, or mental health conditions. Such should be controlled when possible to maximize the strength of the result findings.
Accordingly, the study lacks details on participant engagement with the VR intervention (e.g., feedback, adherence, or session completion rates). These must be further examined to determine whether VR can be a feasible and effective intervention.
It found that the limitations and future directions section should be strengthened. It should discuss limitations related to sample size and generalizability and how this information might be generalizable to other populations or settings.
The association of psychological factors with body image and eating behaviours is certainly difficult to study (since it relies on participants' self-reported experiences); however, citing several recent studies that examine this association enhances the theoretical foundation of this study.
Ju, Q., Wu, X., Li, B., Peng, H., Lippke, S.,... Gan, Y. (2024). Regulation of craving training to support healthy food choices under stress: A randomized control trial employing the hierarchical drift-diffusion model. Applied Psychology: Health and Well-Being, 16(3), 1159-1177. doi: https://doi.org/10.1111/aphw.12522
Zhang, H., Wang, Z., Wang, G., Song, X., Qian, Y., Liao, Z.,... Xia, Y. (2023). Understanding the Connection between Gut Homeostasis and Psychological Stress. The Journal of Nutrition, 153(4), 924-939. doi: https://doi.org/10.1016/j.tjnut.2023.01.026
Luo, S., Yuan, H., Wang, Y., & Bond, M. H. (2024). Culturomics: Taking the cross-scale, interdisciplinary science of culture into the next decade. Neuroscience & Biobehavioral Reviews, 167, 105942. doi: https://doi.org/10.1016/j.neubiorev.2024.105942
Feng, D., Li, P., Xiao, W., Pei, Z., Chen, P., Hu, M.,... Wang, Y. (2023). N6-methyladenosine profiling reveals that Xuefu Zhuyu decoction upregulates METTL14 and BDNF in a rat model of traumatic brain injury. Journal of Ethnopharmacology, 317, 116823. doi: https://doi.org/10.1016/j.jep.2023.116823
Hao, S., Xin, Q., Xiaomin, Z., Jiali, P., Xiaoqin, W., Rong, Y.,... Cenlin, Z. (2023). Group membership modulates the hold-up problem: an event-related potentials and oscillations study. Social Cognitive and Affective Neuroscience, 18(1). doi: 10.1093/scan/nsad071
Comments on the Quality of English Language
The English could be improved to express the research more clearly.
Author Response
|
Response to Reviewer 1 Comments
|
||||
|
1. Summary |
|
|
||
|
Thank you very much for taking the time to review this manuscript. Please find the detailed responses below and the corresponding revisions/corrections highlighted/in track changes in the re-submitted files.
|
||||
|
2. Questions for General Evaluation |
Reviewer’s Evaluation |
Response and Revisions |
||
|
Does the introduction provide sufficient background and include all relevant references? |
Can be improved |
|
||
|
Is the research design appropriate? |
Can be improved |
|
||
|
Are the methods adequately described? |
Can be improved |
|
||
|
Are the results clearly presented? |
Can be improved |
|
||
|
Are the conclusions supported by the results?
|
Can be improved
|
|
||
|
3. Point-by-point response to Comments and Suggestions for Authors |
||||
|
Comments 1: Overly Long and Complicated Title: A more concise title would facilitate a quicker understanding of the study focus.
|
||||
|
Response 1: First, we want to thank Reviewer 1 for the suggestion. Considering your feedback, we propose a more concise title: The role of body image perception on a continuum eating attitudes and behaviors among people seeking treatment and immersed in virtual reality. The manuscript has been revised by modifying the tittle on line 2 to 4 “The role of body image perception on a continuum from dysfunctional to healthy eating attitudes and behaviors among people seeking treatment.”
|
||||
|
Comments 2: The methodology explains that canonical correlation analysis (CCA) is chosen, but it does not explain why CCA was used rather than other methods, such as structural equation modelling (SEM). Explain your reason for this selection, and note any restrictions it may have. |
||||
|
Response 2: In the present article, the main objective is to explore and investigate the relationships between attitudinal and perceptual body image variables, global self-worth, physical self-worth, and physical attractiveness in relation to the continuum from dysfunctional to healthy eating attitudes and behaviors. Canonical correlation analysis (CCA) is an exploratory approach aimed at identifying statistical relationships between two sets of variables, which is perfectly suited for the current context. Our goal is not to confirm the fit of a causal model defined a priori between these variables, to infer causality or test complex models involving mediators or moderators, as is the case in structural equation modeling (SEM). Lu (2019 & 2018) and Petter and Hadavi (2021) argued that CCA directly examines the relationship between two sets of variables, making it straightforward for understanding the interdependencies without the need for the specification of a complex and restrictive model structure required in SEM. However, the use of canonical correlation may present certain limitations, such as considering only observed variables (e.g., total scores on the IES or EDE-Q). For instance, intuitive eating is a concept that can be measured using multiple indicators, and the use of canonical correlation may not fully capture the complexity of some of the underlying variables associated with this concept. The manuscript has been revised by modifying the text on line 232 to 238 “This study used a canonical correlation analysis [36], using SPSS version 29 canonical correlation command and MANOVA command with the discrim option. This analysis follows an exploratory approach, designed to determines the magnitude of the relationship between two multivariate sets of variables referred to as canonical variables. It directly examines the relationship between two sets of variables, which is more straightforward for understanding the interdependencies and avoids the need for specifying of a complex and restrictive model structure required in SEM.”
Comments 3: Insufficient description of the VR intervention Please elaborate on the specific components of VR experiences (e.g., wearing extra weight, what are the targets of emotional and attitudinal aspects of body dissatisfaction) Response 3: Several clarifications have been made regarding the secondary data related to virtual reality. These details enhance understanding of how virtual reality was used in the study. Virtual reality is only used to measure dissatisfaction and body distortion (both allocentric and egocentric). Therefore, we have removed virtual reality from the title of the article to avoid giving the impression that there was an actual VR intervention. However, to keep the text concise, we refer the reader to the study by Monthuy-Blanc (2020) for a more detailed procedure. The current manuscript has been revised by modifying the text on line 186 to 207 ‘’Participants were asked to provide self-reported information on their age, while their current height and weight were reported by caregivers of the eLoriCorps program 1.0 to calculate the BMI. This virtual reality-based adaptation of paper-and-pencil figure rating scales was used to visually estimate body size from both allocentric and egocentric perspectives. It includes three virtual environments: a neutral environment and two spatial frame perspectives featuring male or female body continuum: allocentric and egocentric. Initially, the neutral environment allowed participants to develop the navigation skills necessary to move within the virtual environment and around identical tubular structures, without the presence of virtual bodies. Participants were then asked, from both the allocentric and egocentric perspectives, to select the virtual body that most closely represented their ideal and perceived body size. In the allocentric perspective, users viewed a lineup of seven increasing BMIs, ranging from 15 to 33 kg/m2, or nine virtual bodies (with increasing BMIs from 15 to 40 kg/m2). All virtual bodies were visible in the user's field of vision. To select their perceived and ideal body size, participants walked toward the virtual body that best reflected their perceived and ideal body size. In the egocentric perspective, users were told they would experience the virtual body as if it were their own, looking down from the chest. The user could move their head along three degrees of freedom (yaw, pitch, and roll). The experimenter then explained that users would experience each virtual body, from the thinnest to the largest. To select their perceived and ideal body size, participants guided the experimenter to the virtual body that most closely matched their ideal and perceived size’’. Considering the recent additions to the description of how virtual reality is used, do you think it would be more appropriate to include the following statement—'Virtual reality allows for the exploration of the perceptual-sensory-affective dimensions of body dissatisfaction, with particular emphasis on the sensory dimension from an egocentric perspective'—in the discussion or in the section on the study’s advantages, rather than in the assessment measures?
Comments 4: The results section should contain specific statistical data (e.g., correlation coefficients, p-values, etc.) to adequately support the author's claims about the relationships between body image and eating behaviours.
Response 4: The manuscript already included statistical data such as correlations and p values. However, we concur with the reviewer for the need to provide more information to support the claims. We added the correlations between the attitudinal and perceptual variables and the canonical variable representing the continuum of EAB. The manuscript has been revised by modifying the text on lines 291 to 317 “. The correlation between the variables measuring attitudinal and perceptual dimensions of body image and the canonical scores of the continuum from dysfunctional to healthy eating attitudes and behaviors (i.e., cross-loadings) were, respectively, - .18 (p = .06, ns) for body distortion from the egocentric perspective, -.21 (p = .023) for body distortion from the allocentric perspective, .43 (p < .001) for body dissatisfaction from the egocentric perspective, .48 (p < .001) for body dissatisfaction from the allocentric perspective, .65 (p < 001) for the Symptom index of the EDI, -.59 (p < .001) for the Global self-worth subscale of the PSI, -0.45 (p < .001) for the Physical self-worth subscale of the PSI, and -.54 (p < .001) for the Physical attractiveness Subscale of the PSI. Figure 2 displays simultaneously the scatterplots of the two correlations between the canonical variate representing the set of attitudinal and perceptual variables (i.e., Set 2) and the weighted scores (i.e., after applying the derived canonical weights to the original variables) of the variables constituting the continuum from healthy (as measured by the IES) to dysfunctional (as measured by the EDEQ) eating. It shows the direction and strengths of the correlations, but most importantly that attitudinal and perceptual variables were contributing differently at both ends of the continuum. These results support the interest of conceptualizing EAB along a continuum from healthy to dysfunctional eating.”
Response 5: The results did not differ when analysed separately for each sex (i.e.., significant differences remained significant and non-significant differences remained non-significant). Also, performing the canonical correlation analysis with the inclusion of age did not change the results, as age did not significantly contribute to the model and other loading remained similar. However, the addition of age reduced the power of the statistical analyses. Finaly, the role of mental health conditions could not be tested as confounder, because: (a) mental health conditions other than eating disorders, and (b) the dysfunctionality of eating disorders is already included in the first set of variables. The manuscript has been revised by adding to the methodological section below sentences, on line 246 to 252 “To document the potential impact of sex on the results, all statistical analyses were also performed separately for females and males. The results did not differ when analyzed separately for each sex (i.e.., significant differences remained significant and non-significant differences remained non-significant). Including age in the analysis also did not change the results and the contribution of age was non-significant. Therefore, to maximize statistical power, results for sex and age are not reported (analyses by sex and with age are available upon request).”
Comments 6: Accordingly, the study lacks details on participant engagement with the VR intervention (e.g., feedback, adherence, or session completion rates). These must be further examined to determine whether VR can be a feasible and effective intervention. Response 6: Thank you for sharing this comment. In this study, VR was not used for an intervention. It was only sed to measure body image distortion and dissatisfaction from the allocentric and egocentric perspectives. This impression was probably created by the mention of VR in the title of the article. We have therefore removed reference to VR in the title.
Comments 7: It found that the limitations and future directions section should be strengthened. It should discuss limitations related to sample size and generalizability and how this information might be generalizable to other populations or settings. Response 7: This is an interesting comment, and we investigated as you requested. Indeed, several limitations of the current study need to be addressed, along with future directions. The manuscript has been revised by adding to the Strengths and Limitations section below sentences, on line 426 to 441 “Regarding the sample size, it would be valuable for future studies to explore the robustness of the associations and ensure better generalizability of the results by using a larger and more heterogeneous sample relatively to sexes and gender. Considering that the statistics only compared sexes and given that the majority of the study sample consisted of women (with only 10 men), the findings may not be applicable to individuals of other genders, as body image perceptions can vary across genders mainly and sexes [59,60]. Future research should aim to include a more balanced representation of male and female sample to investigate potential gender-based differences. Indeed, it may be valuable to include certain sociodemographic variables that were not measured in the present study but could have an impact, such as sexual orientation or social support [61]. Additionally, while covering a wide age range (13 to 80 years) can be informative, it may affect the generalizability of the results, particularly for less-represented age groups. Similarly, although having a broad BMI spectrum is valuable, it can introduce biases in the interpretation of the results related to eating behaviors and body image perceptions, as low BMIs may be more associated with restrictive eating behaviors, while higher BMIs may be more linked to binge eating [62,63].”
Comments 8: The association of psychological factors with body image and eating behaviours is certainly difficult to study (since it relies on participants' self-reported experiences); however, citing several recent studies that examine this association enhances the theoretical foundation of this study. Response 8: Thank you for this positive comments. Indeed, you are right to highlight the challenges and issues related to the use of self-reported measures, as well as the benefits of incorporating other methods to support theoretical models. Although self-reported measures have certain limitations, they remain an important source for understanding individuals' subjective experiences and perceptions of body image and eating behaviors. In fact, many studies use this type of method to measure the concept of body image perceptions. Furthermore, virtual reality represents an intriguing avenue for capturing the complexity of body image perceptions and exploring dimensions that are only accessible through this technology (e.g., sensory dimensions). While other studies using biological and neurobiological markers are valuable, the present study aims to complement these approaches by focusing more on psychological and perceptual markers.
|
||||
|
4. Response to Comments on the Quality of English Language |
||||
|
Point 1: Can be improved |
||||
|
Response 1: Track changes has not been used here for readability purposes.
|
||||
Reviewer 2 Report
Comments and Suggestions for Authors
Thank you for the invitation to review the manuscript titled "The role of the perceptual and attitudinal dimensions of body image perception on a continuum from dysfunctional to healthy eating attitudes and behaviors among people seeking treatment and immersed in virtual reality." Overall, the manuscript is well-written; however, for its publication, the authors need to address the following issues:
- Could you provide more details about the inclusion and exclusion criteria for participants? How were they screened and recruited for the study?
- Were there any demographic variables (e.g., socioeconomic status, education level) that were considered during participant selection, and could these have influenced the results?
- Could you elaborate on the specific features of the app (e.g., type of activities, frequency of prompts) and how they were tailored to address anxiety and depression?
- Which validation processes were conducted to ensure that the tools used for measuring anxiety (GAD-7), depression (PHQ-9), and wellbeing (WHO-5) were appropriate for the target population?
- What criteria were used to determine the clinical significance of the reductions in GAD-7 and PHQ-9 scores? Are these reductions sufficient to suggest meaningful improvement in participants' daily lives?
- Could you discuss more explicitly the potential advantages or limitations of your study?
- How might the voluntary nature of participation and high adherence rates impact the generalizability of your findings? Could participants' motivation levels have biased the results?
Author Response
|
Response to Reviewer 2 Comments
|
||
|
1. Summary |
|
|
|
Thank you very much for taking the time to review this manuscript. Please find the detailed responses below and the corresponding revisions/corrections highlighted/in track changes in the re-submitted files.
|
||
|
2. Questions for General Evaluation |
Reviewer’s Evaluation |
Response and Revisions |
|
Does the introduction provide sufficient background and include all relevant references? |
Can be improved |
|
|
Is the research design appropriate? |
Can be improved |
|
|
Are the methods adequately described? |
Can be improved |
|
|
Are the results clearly presented? |
Can be improved |
|
|
Are the conclusions supported by the results?
|
Can be improved
|
|
|
3. Point-by-point response to Comments and Suggestions for Authors |
||
|
Comments 1: Could you provide more details about the inclusion and exclusion criteria for participants? How were they screened and recruited for the study?
|
||
|
Response 1: The manuscript already reports details about the inclusion and exclusion criteria on lines 155 to 161, and we are happy to provide you more details. This retrospective study was conducted by analyzing the anonymized secondary data collected in the Loricorps database. The Loricorps database a secured data platform (LDB) used in this study were collected during enrollment to the eLoriCorps Program 1.0 (adapted version from Monthuy-Blanc et al., 2016), an e-Health transdisciplinary program, delivered at the university-based clinic, between September 2017 and December 2022. The eLoriCorps Program 1.0 addresses dysfunctional eating attitudes and behaviors resulting from perceptual disorders of all ages and forms, sub-clinical or clinical (eating disorders, sports eating disorders, etc.), of mild to moderate severity. The exclusion criteria were the presence of a severe self-reported comorbid psychiatric conditions (e.g., personality disorders, psychosis or severe anxiety or de-pression). The manuscript has been revised by adding to the Data sources section below sentences, on line 158 to 162 “The eating disorder-related data were collected by healthcare providers during sessions conducted in accordance with the intervention protocol of the eLoriCorps Program 1.0. Comorbidities were assessed by the transdisciplinary clinical team of the eLoriCorps Program 1.0 and confirmed by a diagnostic specialist (e.g. psychologist) using DSM-5 criteria”.
|
||
|
Comments 2: Were there any demographic variables (e.g., socioeconomic status, education level) that were considered during participant selection, and could these have influenced the results? |
||
|
Response 2: As already mentioned in Table 1, some sociodemographic variables have been documented. The potential role of sex and age has been also raised by Reviewer 1. These two variables did not influence the results, as can be seen on page 11. The manuscript has been revised by adding to the Strengths and Limitations section below sentences, on line 433 to 435 “Indeed, it may be valuable to include certain sociodemographic variables that were not measured in the present study but could have an impact, such as sexual orientation or social support.”
Comments 3: Could you elaborate on the specific features of the app (e.g., type of activities, frequency of prompts) and how they were tailored to address anxiety and depression? Response 3: It is difficult to answer this comment raised by the reviewer, because there is no app used in the study. The article does not mention an app and its features. In addition, the study did not target anxiety and depression. We remain open to provide additional information, if the reviewer can clarify what he or she is referring to. Maybe the confusion stems from the mention of VR in the title. Because VR was used only to assess the allo- and ego-centric perspectives, we removed it from the title.
Comments 4: Which validation processes were conducted to ensure that the tools used for measuring anxiety (GAD-7), depression (PHQ-9), and wellbeing (WHO-5) were appropriate for the target population?
Response 4: It is difficult to answer this comment raised by the reviewer, because we did not use the GAD-7, the PHQ-9 and the WHO-5. There is no mention of these measures in the manuscript. We remain open to provide additional information, if the reviewer can clarify what he or she is referring to.
Response 5: It is difficult to answer this comment raised by the reviewer, because the current study did not use the GAD-7 of the PHQ-9. There is no mention of these instruments in the manuscript. In addition, there is no intervention mentioned in the manuscript that could lead to improvements in participants daily lives. We remain open to provide additional information, if the reviewer can clarify what he or she is referring to.
Comments 6: Could you discuss more explicitly the potential advantages or limitations of your study? Response 6: As recommended by Reviewer 1 and 3, we added more information about the potential limitations of the study. Indeed, several limitations of the current study need to be addressed, along with future directions. The manuscript has been revised by adding to the Strengths and Limitations section, on line 417 to 441: “Nevertheless, considering the sample, we find it valuable to report results from individuals with DEAB with mild to moderate severity. It provides information on a large portion of the population. As mentioned in the manuscript, up to 48% of individuals may exhibit disordered eating and eating disorders when conceptualized along a continuum of eating attitudes and behaviors. In future research, it is worthwhile to focus on more severe cases in order to better understand the relationship between eating behaviors and body image perceptions within this specific population. It should be noted, however, that the current study is not without limitations. Regarding the sample size, it would be valuable for future studies to explore the robustness of the associations and ensure better generalizability of the results by using a larger and more heterogeneous sample relatively to sexes and gender. Considering that the statistics only compared sexes and given that the majority of the study sample consisted of women (with only 10 men), the findings may not be applicable to individuals of other genders, as body image perceptions can vary across genders mainly and sexes [59,60]. Future research should aim to include a more balanced representation of male and female sample to investigate potential gender-based differences. Indeed, it may be valuable to include certain sociodemographic variables that were not measured in the present study but could have an impact, such as sexual orientation or social support [61]. Additionally, while covering a wide age range (13 to 80 years) can be informative, it may affect the generalizability of the results, particularly for less-represented age groups. Similarly, although having a broad BMI spectrum is valuable, it can introduce biases in the interpretation of the results related to eating behaviors and body image perceptions, as low BMIs may be more associated with restrictive eating behaviors, while higher BMIs may be more linked to binge eating [62,63].”
Comments 7: How might the voluntary nature of participation and high adherence rates impact the generalizability of your findings? Could participants' motivation levels have biased the results? Response 7: Indeed, the results may be influences by the voluntary nature of the participation. Eating behaviors and body image perceptions remain sensitive topics to address. Some individuals may be ambivalent about changing their eating behaviors, which can lead them to either minimize or exaggerate certain aspects of their eating habits. However, there is no adherence rates, as no treatment was provided.
|
||
Reviewer 3 Report
Comments and Suggestions for Authors
I would like to express my appreciation for the opportunity to review this manuscript. The study's objective is both interesting and of significant importance. I commend the author(s) for their investigation. The study provides an insightful examination into the psychological factors influencing eating behavior through the lens of body image perception. The study, conducted within a virtual reality framework, explores the complex interaction between body image, from both allocentric and egocentric viewpoints, and eating behaviors along a continuum from functional to dysfunctional. In my opinion, the manuscript requires revisions:
1. The use of different rating scales (7-point and 9-point) in assessments could lead to inconsistencies in how participants perceive and respond to these scales, potentially introducing response biases that could affect the study’s conclusions. To address the inconsistency and potential response bias introduced by using different rating scales you can implement Z-Score Standardization or IRT models or Composite Score Formation. The most beneficial in my opinion is to use a combination of methods.
2. The participants were individuals seeking treatment and may not represent the broader population. This selection could distort results towards more severe cases of body image disturbance and eating disorders.
In conclusion, despite some reservations, the study's results pave the way for new directions in research and clinical applications, especially in crafting virtual reality-based interventions for eating disorders.
Author Response
Response to Reviewer 3 Comments
|
1. Summary |
|
|
|
Thank you very much for taking the time to review this manuscript. Please find the detailed responses below and the corresponding revisions/corrections highlighted/in track changes in the re-submitted files.
|
||
|
2. Questions for General Evaluation |
Reviewer’s Evaluation |
Response and Revisions |
|
Does the introduction provide sufficient background and include all relevant references? |
Yes |
|
|
Is the research design appropriate? |
Can be improved |
|
|
Are the methods adequately described? |
Can be improved |
|
|
Are the results clearly presented? |
Yes |
|
|
Are the conclusions supported by the results?
|
Yes
|
|
|
3. Point-by-point response to Comments and Suggestions for Authors |
||
|
Comments 1: The use of different rating scales (7-point and 9-point) in assessments could lead to inconsistencies in how participants perceive and respond to these scales, potentially introducing response biases that could affect the study’s conclusions. To address the inconsistency and potential response bias introduced by using different rating scales you can implement Z-Score Standardization or IRT models or Composite Score Formation. The most beneficial in my opinion is to use a combination of methods.
|
||
|
Response 1: First, we want to thank Reviewer 3 the Comments and Suggestions. As mentioned on lines 208 to 212, Z-scores were indeed calculated for the measures of ideal and perceived body sizes, because different rating scales were used (i.e., a 7-point and a 9-point scale), including Z ideal body size – Allocentric, Z Ideal body size Egocentric, Z Perceived body size Egocentric, Z Perceived body size Allocentric (for a detailed description of the procedure please refer to Monthuy-Blanc et al., 2022) [27]. Note that, in the article we only report standardized canonical coefficients, which essentially represent the information as if all scales had been rescaled like a Z score. In the article, Item-Response Theory transformations cannot be used here, because the constructs are measured using Likert scale and not difficulty levels of each item. Composite Score Formation cannot be used in this article, because there is no a priori justification to guide how to combine and weight the different variables. The advantage of the canonical correlation analysis is to reveal the relevance and contribution of each variable to their respective set. Actually, our finding now provides the information required to create composite scores.
|
||
|
Comments 2: The participants were individuals seeking treatment and may not represent the broader population. This selection could distort results towards more severe cases of body image disturbance and eating disorders.
Response 2: Indeed, you are right to mention that the results may not be generalizable to populations with severe and chronic levels of DEAB, or to individuals who are not seeking intervention. This observation has been added in the revised manuscript, in the Strengths and Limitations section on line 417 to 424 “Nevertheless, we find it valuable to report results from individuals with subclinical or clinically DEAB with mild to moderate severity. It provides information on a large portion of the population. As mentioned in the manuscript, up to 48% of individuals may exhibit disordered eating and eating disorders when conceptualized along a continuum of eating attitudes and behaviors. In future research, it is worthwhile to focus on more severe cases in order to better understand the relationship between eating behaviors and body image perceptions within this specific population.”
4. Response to Comments on the Quality of English Language Response 1: Track changes has not been used here for readability purposes.
5. Additional clarifications The changes made to the manuscript have been highlighted in yellow.
|
||
Round 2
Reviewer 1 Report
Comments and Suggestions for Authors
Please revise the manuscript according to the suggestions previously provided. Ensure that all comments are read carefully and addressed thoroughly in the revised draft. Additionally, please update the literature review using the citations provided. These revisions must be made to enhance the clarity and depth of the manuscript.
Ju, Q., Wu, X., Li, B., Peng, H., Lippke, S.,... Gan, Y. (2024). Regulation of craving training to support healthy food choices under stress: A randomized control trial employing the hierarchical drift-diffusion model. Applied Psychology: Health and Well-Being, 16(3), 1159-1177. doi: https://doi.org/10.1111/aphw.12522
Zhang, H., Wang, Z., Wang, G., Song, X., Qian, Y., Liao, Z.,... Xia, Y. (2023). Understanding the Connection between Gut Homeostasis and Psychological Stress. The Journal of Nutrition, 153(4), 924-939. doi: https://doi.org/10.1016/j.tjnut.2023.01.026
Luo, S., Yuan, H., Wang, Y., & Bond, M. H. (2024). Culturomics: Taking the cross-scale, interdisciplinary science of culture into the next decade. Neuroscience & Biobehavioral Reviews, 167, 105942. doi: https://doi.org/10.1016/j.neubiorev.2024.105942
Feng, D., Li, P., Xiao, W., Pei, Z., Chen, P., Hu, M.,... Wang, Y. (2023). N6-methyladenosine profiling reveals that Xuefu Zhuyu decoction upregulates METTL14 and BDNF in a rat model of traumatic brain injury. Journal of Ethnopharmacology, 317, 116823. doi: https://doi.org/10.1016/j.jep.2023.116823
Hao, S., Xin, Q., Xiaomin, Z., Jiali, P., Xiaoqin, W., Rong, Y.,... Cenlin, Z. (2023). Group membership modulates the hold-up problem: an event-related potentials and oscillations study. Social Cognitive and Affective Neuroscience, 18(1). doi: 10.1093/scan/nsad071
Comments on the Quality of English LanguageThe English could be improved to express the research more clearly.
Author Response
|
Response to Reviewer 1 Comments- Round 2
|
||||||||||||||||||||||||
|
1. Summary |
|
|
||||||||||||||||||||||
|
Thank you very much for taking the time to review this manuscript. Please find the detailed responses below and the corresponding revisions/corrections highlighted/in track changes in the re-submitted files.
|
||||||||||||||||||||||||
|
2. Questions for General Evaluation |
Reviewer’s Evaluation |
Response and Revisions |
||||||||||||||||||||||
|
Does the introduction provide sufficient background and include all relevant references? |
Must be improved |
|
||||||||||||||||||||||
|
Is the research design appropriate? |
Can be improved |
|
||||||||||||||||||||||
|
Are the methods adequately described? |
Can be improved |
|
||||||||||||||||||||||
|
Are the results clearly presented? |
Can be improved |
|
||||||||||||||||||||||
|
Are the conclusions supported by the results?
|
Can be improved
|
|
||||||||||||||||||||||
|
3. Point-by-point response to Comments and Suggestions for Authors General comments: Please revise the manuscript according to the suggestions previously provided. Ensure that all comments are read carefully and addressed thoroughly in the revised draft. |
||||||||||||||||||||||||
|
Comments 1: Overly Long and Complicated Title: A more concise title would facilitate a quicker understanding of the study focus.
|
||||||||||||||||||||||||
|
Response 1: First, we want to thank Reviewer 1 for the suggestion. Considering your feedback, we propose a more concise title: The role of body image perception on a continuum from dysfunctional to healthy eating attitudes and behaviors among people seeking treatment. The manuscript has been revised by modifying the tittle on line 2 to 4 “The role of body image perception on a continuum from dysfunctional to healthy eating attitudes and behaviors among people seeking treatment.”
|
||||||||||||||||||||||||
|
Comments 2: The methodology explains that canonical correlation analysis (CCA) is chosen, but it does not explain why CCA was used rather than other methods, such as structural equation modelling (SEM). Explain your reason for this selection, and note any restrictions it may have. |
||||||||||||||||||||||||
|
Response 2: In the present article, the main objective is to explore and investigate the relationships between attitudinal and perceptual body image variables, global self-worth, physical self-worth, and physical attractiveness in relation to the continuum from dysfunctional to healthy eating attitudes and behaviors. Canonical correlation analysis (CCA) is an exploratory approach aimed at identifying statistical relationships between two sets of variables, which is perfectly suited for the current context. Our goal is not to confirm the fit of a causal model defined a priori between these variables, to infer causality or test complex models involving mediators or moderators, as is the case in structural equation modeling (SEM). Lu (2019 & 2018) and Petter and Hadavi (2021) argued that CCA directly examines the relationship between two sets of variables, making it straightforward for understanding the interdependencies without the need for the specification of a complex and restrictive model structure required in SEM. However, the use of canonical correlation may present certain limitations, such as considering only observed variables (e.g., total scores on the IES or EDE-Q). For instance, intuitive eating is a concept that can be measured using multiple indicators, and the use of canonical correlation may not fully capture the complexity of some of the underlying variables associated with this concept. The manuscript has been revised by modifying the text on line 231 to 237 “This study used a canonical correlation analysis [37], using SPSS version 29 canonical correlation command and MANOVA command with the discriminate option. This analysis follows an exploratory approach, designed to determine the magnitude of the relationship between two multivariate sets of variables referred to as canonical variables. It directly examines the relationship between two sets of variables, which is more straightforward for understanding the interdependencies and avoids the need for specifying of a complex and restrictive model structure required in SEM.”
Comments 3: Insufficient description of the VR intervention Please elaborate on the specific components of VR experiences (e.g., wearing extra weight, what are the targets of emotional and attitudinal aspects of body dissatisfaction) Response 3: Several clarifications have been made regarding the secondary data related to virtual reality. These details enhance understanding of how virtual reality was used in the study. Virtual reality is only used to measure dissatisfaction and body distortion (both allocentric and egocentric). Therefore, we have removed virtual reality from the title of the article to avoid giving the impression that there was an actual VR intervention. However, to keep the text concise, we refer the reader to the study by Monthuy-Blanc (2022) for a more detailed procedure. The current manuscript has been revised by modifying the text on line 186 to 206 ‘’Participants were asked to self-report their age, while their current height and weight were reported by caregivers using the eLoriCorps 1.0 program to calculate BMI. This virtual reality-based adaptation of paper-and-pencil figure rating scales was used to visually estimate body size from both allocentric and egocentric perspectives. It includes three virtual environments: a neutral environment and two spatial frame perspectives with a male or female body continuum: allocentric and egocentric. First, the neutral environment allowed participants to develop the navigation skills necessary to move within the virtual environment and around identical tubular structures, without the presence of virtual bodies. Participants were then asked, from both the al-allocentric and egocentric perspectives, to select the virtual body that most closely represented their ideal and perceived body size. In the allocentric perspective, users viewed a lineup of seven increasing BMIs, ranging from 15 to 33 kg/m2, or nine virtual bodies (with increasing BMIs ranging from 15 to 40 kg/m2). All virtual bodies were visible in the user's field of vision. To select their perceived and ideal body size, participants walked toward the virtual body that best reflected their perceived and ideal body size. In the egocentric perspective, users were told that they would experience the virtual body as if it were their own, looking down from the chest. The user could move their head along three degrees of freedom (yaw, pitch, and roll). The experimenter then explained that users would experience each virtual body, from the thinnest to the largest. To select their perceived and ideal body size, participants guided the experimenter to the virtual body that most closely matched their ideal and perceived size’’. Considering the recent additions to the description of how virtual reality is used, do you think it would be more appropriate to include the following statement—'Virtual reality allows for the exploration of the perceptual-sensory-affective dimensions of body dissatisfaction, with particular emphasis on the sensory dimension from an egocentric perspective'—in the discussion or in the section on the study’s advantages, rather than in the assessment measures?
Comments 4: The results section should contain specific statistical data (e.g., correlation coefficients, p-values, etc.) to adequately support the author's claims about the relationships between body image and eating behaviours.
Response 4: The manuscript already included statistical data such as correlations and p values. However, we concur with the reviewer for the need to provide more information to support the claims. We added the correlations between the attitudinal and perceptual variables and the canonical variable representing the continuum of EAB. The manuscript has been revised by modifying the text on lines 290 to 325 “. The correlation between the variables measuring attitudinal and perceptual dimensions of body image and the canonical scores of the continuum from dysfunctional to healthy eating attitudes and behaviors (i.e., cross-loadings) were, respectively, - .18 (p = .06, ns) for body distortion from the egocentric perspective, -.21 (p = .023) for body distortion from the allocentric perspective, .43 (p < .001) for body dissatisfaction from the egocentric perspective, .48 (p < .001) for body dissatisfaction from the allocentric perspective, .65 (p < 001) for the Symptom index of the EDI, -.59 (p < .001) for the Global self-worth subscale of the PSI, -0.45 (p < .001) for the Physical self-worth subscale of the PSI, and -.54 (p < .001) for the Physical attractiveness Subscale of the PSI. Figure 2 displays simultaneously the scatterplots of the two correlations between the canonical variate representing the set of attitudinal and perceptual variables (i.e., Set 2) and the weighted scores (i.e., after applying the derived canonical weights to the original variables) of the variables constituting the continuum from healthy (as measured by the IES) to dysfunctional (as measured by the EDEQ) eating. It shows the direction and strengths of the correlations, but most importantly that attitudinal and perceptual variables were contributing differently at both ends of the continuum. These results support the interest of conceptualizing EAB along a continuum from healthy to dysfunctional eating.”
Response 5: The results did not differ when analyzed separately for each sex (i.e.., significant differences remained significant and non-significant differences remained non-significant). Also, performing the canonical correlation analysis with the inclusion of age did not change the results, as age did not significantly contribute to the model and other loading remained similar. However, the addition of age reduced the power of the statistical analyses. Finaly, the role of mental health conditions could not be tested as confounder, because: (a) mental health conditions other than eating disorders, and (b) the dysfunctionality of eating disorders is already included in the first set of variables. The manuscript has been revised by adding to the methodological section below sentences, on line 245 to 251 “To document the potential impact of sex on the results, all statistical analyses were also performed separately for females and males. The results did not differ when analyzed separately for each sex (i.e.., significant differences remained significant and non-significant differences remained non-significant). Including age in the analysis also did not change the results and the contribution of age was non-significant. Therefore, to maximize statistical power, results for sex and age are not reported (analyses by sex and with age are available upon request).”
Comments 6: Accordingly, the study lacks details on participant engagement with the VR intervention (e.g., feedback, adherence, or session completion rates). These must be further examined to determine whether VR can be a feasible and effective intervention. Response 6: Thank you for sharing this comment. In this study, VR was not used for an intervention. It was only sed to measure body image distortion and dissatisfaction from the allocentric and egocentric perspectives. This impression was probably created by the mention of VR in the title of the article. We have therefore removed reference to VR in the title.
Comments 7: It found that the limitations and future directions section should be strengthened. It should discuss limitations related to sample size and generalizability and how this information might be generalizable to other populations or settings. Response 7: This is an insightful comment raised by you and Reviewer 3, and we have thoroughly investigated it as requested. Indeed, it is essential to address several strengths and limitations of the current study, as well as outline potential directions for future research. The manuscript has been revised by adding to the Strengths and Limitations section below sentences, on line 420 to 449 “Nevertheless, given the sample, we find it valuable to report results from individuals with DEAB with mild to moderate severity. It provides information about a large portion of the population. As mentioned in the manuscript, up to 48% of individuals may exhibit disordered eating and eating disorders when conceptualized along a continuum-um of eating attitudes and behaviors. In future research, it is worthwhile to focus on more severe cases in order to better understand the relationship between eating behaviors and body image perceptions within this specific population. It should be noted, however, that the current study is not without limitations. In terms of sample size, it would be valuable for future studies to examine the robustness of the associations and ensure better generalizability of the results by using a larger and more heterogeneous sample in terms of sexes and gender. Given that the statistics only compared sexes and given that the majority of the study sample consist-ed of women (with only 10 men), the findings may not be applicable to individuals of other genders, as body image perceptions can vary across genders mainly and sexes [61,62]. Future research should aim to include a more balanced representation of male and female samples to investigate potential gender-based differences. Indeed, it may be valuable to include certain sociodemographic variables that were not measured in the present study but could have an impact, such as sexual orientation, culture, or social support [63]. Furthermore, given that body dissatisfaction varies across different cultures, the inclusion of this variable in future research could provide additional in-sights and enhance the study’s relevance [51]. A culturomics approach could provide valuable insights into investigating EAB and the cultural representations of body im-age perceptions, spanning from micro to macro levels [64]. In addition, although the inclusion of a broad age range (13 to 80 years) can be in-formative, it may affect the generalizability of the results, particularly for less-represented age groups. Similarly, a wide range of BMIs, although valuable, may introduce bias in the interpretation of results related to eating behaviors and body image perceptions, as low BMIs may be more associated with restrictive eating behaviors, whereas higher BMIs may be more associated with binge eating [65,66].”
Comments 8: The association of psychological factors with body image and eating behaviours is certainly difficult to study (since it relies on participants' self-reported experiences); however, citing several recent studies that examine this association enhances the theoretical foundation of this study. Response 8: Thank you for these comments. Indeed, you are right to highlight the challenges and issues related to the use of self-reported measures, as well as the benefits of incorporating other methods to support theoretical models. Although self-reported measures have certain limitations, they remain an important source for understanding individuals' subjective experiences and perceptions of body image and eating behaviors. In fact, many studies use this type of method to measure the concept of body image perceptions. Furthermore, virtual reality represents an intriguing avenue for capturing the complexity of body image perceptions and exploring dimensions that are only accessible through this technology (e.g., sensory dimensions). While other studies using biological and neurobiological markers are valuable, the present study aims to complement these approaches by focusing more on psychological and perceptual markers.
Comment 9: Please update the literature review using the citations provided. These revisions must be made to enhance the clarity and depth of the manuscript. Response 9: Thank you for your suggestion. After careful examination as detailed in Table 1 below, the relevance of all suggested references must be seriously questioned, and more focused reference could be used to illustrate the clinical and cultural implications of our findings. We followed the intentions of the reviewer and found a reference that is even more specific to our study than the one proposed and conserved one of them. We have therefore incorporated the newly identified reference along with one of the proposed references, and we sincerely thank the reviewer for their valuable suggestion. The manuscript has been revised by adding to the Discussion section below sentences, on line 380 to 383 “Cognitive-behavioral therapy (CBT) or culturally adapted cognitive-behavioral techniques that integrate culturally relevant values and norms, combined with innovative digital and technological interventions such as virtual reality, have shown great promise in understanding and addressing these challenges. [50,51]’’ and on line 408 to 411 “Furthermore, in stressful situations, combining strategies that address both food choices and stress management (such as cognitive interventions) may be promising tools for promoting healthier and more functional food choices [60] ’’ and Strengths and Limitations section below sentences, on line 438 to 443 “Indeed, it may be valuable to include certain sociodemographic variables that were not measured in the present study but could have an impact, such as sexual orientation, culture, or social support [63]. Furthermore, given that body dissatisfaction varies across different cultures, the inclusion of this variable in future research could provide additional insights and enhance the study’s relevance [51]. A culturomics approach could provide valuable insights into investigating EAB and the cultural representations of body image perceptions, spanning from micro to macro levels [64]”. |
||||||||||||||||||||||||
|
|
||||||||||||||||||||||||
|
|
||||||||||||||||||||||||
|
|
||||||||||||||||||||||||